Journal of
**open** psychology data

# Data on Connection With the Natural Environment and Its Impact on Mental Health Among Allotment and Non-Allotment Owners

**DATA PAPER**

**LEANNE HAYWOOD** (iD)

**JAMES STILLER** (iD)

*Author affiliations can be found in the back matter of this article

]u[ ubiquity press

## ABSTRACT

This data file contains demographic information along with established scale scores of 515 both allotment and non-allotment owners to explore their time in nature and the impacts of this on mental health. Questions were also asked around time and activities in nature. The age range was between 18–85 years old. The data had approval from the research ethics committee. Before data collection began, full informed consent was obtained from participants as well as information sheets provided to them and all participants were debriefed afterwards. Identifying information was removed to ensure confidentiality and anonymity. The scores for scales used are included.

**CORRESPONDING AUTHOR:**
**Leanne Haywood**

University of Chichester, UK

leannehaywood95@hotmail.co.uk

**KEYWORDS:**
Nature; Wellbeing; Mental Health; Time in nature; connection to nature

**TO CITE THIS ARTICLE:**
Haywood, L., & Stiller, J. (2024). Data on Connection With the Natural Environment and Its Impact on Mental Health Among Allotment and Non-Allotment Owners. *Journal of Open Psychology Data,* 12: 9, pp. 1–7. https://doi.org/10.5334/jopd.122

# (1) BACKGROUND

Research has shown that spending time in nature has an important impact on wellbeing (Russell et al., 2013). Sessions such as gardening have been shown to encourage rehabilitation from mental and physical health conditions (Martin et al., 2020; Rappe, Koivunen & Korpela, 2008). Even just exposure to the natural environment has significant impacts on health and has been shown to increase positive emotions (Mayer, Frantz, Bruehlman-Senecal & Dolliver, 2009). Often, people who live in urban/suburban environments can become disconnected from nature, this highlights the need for a connection to nature which has shown to have positive impacts on an individual's mental health. This data set aims to develop the current knowledge by providing a large quantitative data set exploring connection to nature and impacts on wellbeing.

Research into mental health has shown there is still more information needed to assess the benefits of being in nature, especially with large samples (Windhorst & Williams, 2015). Physical health has also been shown to have effects on self-esteem (Elavsky et al., 2005; Logi Kristjánsson, Dóra Sigfúsdóttir & Allegrante, 2010), which could be examined further by assessing self-esteem impacts from physical activity in nature. Research has also shown that taking part in gardening activities can have significant improvements on self-esteem (D'Andrea, Batavia & Sasson, 2007; Wood, Pretty & Griffin, 2016). The wealth of previous research on the topic suggests certain groups of people, such as those who garden or spend more time in nature, can accrue psychological benefits. However, more research is needed to tie together these different strands, to see to what extent these populations are affected and what the actual psychological benefits are.

## RATIONALE AND AIMS OF THE CURRENT STUDY

This research study aimed to look at various predictors of mental health in terms of their relation to nature. Taking part in physical activity in nature has been shown to have significant improvements on mental wellbeing (Network, 2017; Paluska & Schwenk, 2000; Penedo & Dahn, 2005), therefore this measure was important to consider in the present study. As it is important to assess this variable as taking part in physical activity in nature could be more effective than just purely accessing nature, as research has suggested this to be the case (Shanahan et al., 2015; Shanahan et al., 2016). Self-esteem has also been shown to be an important factor in improving mental health with previous research showing it is a predictive factor in both good physical and mental health. It has also been shown that nature alone seems to have significant positive impacts on self-esteem, and taking part in physical activity in nature significantly improves both self-esteem and mental health (Barton & Pretty, 2010; Mann, Hosman, Schaalma & De Vries, 2004; Pretty, Peacock, Sellens & Griffin, 2005). This shows the importance of self-esteem

in mental health and connection to nature and so seems to be an important factor when researching nature and wellbeing. Social support similarly, was shown to be effective in improving mental health and was positively affected when spending regular time in natural environments, especially in activities such as gardening/allotments and community-based nature activities. It was also shown that those who spend less time in nature report feeling lower levels of social support (Bloom, 1990; Thoits, 2011), this was important to explore in the current study to see if spending time in nature improves social support, this can in turn increase wellbeing. Finally, connection to nature is important to investigate, as having a strong connection to the natural environment has been reported to prevent mental ill-health (Bratman, Daily, Levy & Gross, 2015; Maller, Townsend, Pryor, Brown & St Leger, 2006; Richardson et al., 2021).

The research expanded on existing theoretical models (Cervinka, Röderer & Hefler, 2012; Luck, Davidson, Boxall & Smallbone, 2011) which focused on human wellbeing and connection to nature, by including measures on self-esteem and social support (Barton et al., 2016; Greenleaf & Roessger, 2017; Rappe, Koivunen & Korpela, 2008), as these factors have been found across the literature to show improvements when exposed to nature, but have not been investigated together. For the current study, mental health, self-esteem, physical health, connection to nature, social identity, loneliness, social support and self-efficacy were explored. This was because these factors were common in the literature but are rarely looked at together in relation to allotment and non-allotment owners. It was hypothesised that increases in self-esteem, social support, connection to nature and physical health will lead to increased mental health. It was also hypothesised that social support and self-esteem will significantly increase in individuals who have a high link to nature (through contact time or connection), such as those who work on allotments.

It was important in the current study to gain insight into how participants engage with nature and how this impacts on wellbeing/mental health. In particular:

1. Does the number of hours spent in nature have a positive effect on mental health and are allotment owners more likely to participate in a wider range of activities in nature?
2. Does self-esteem, physical health, social support, connection to nature and hours spent in nature impact mental wellbeing?

# (2) METHODS

## 2.1 STUDY DESIGN

An online survey design was used to look at the differences between allotment and non-allotment owners and the relationship between mental health,

self-esteem, physical health, connection to nature, social identity, loneliness, social support and self-efficacy The outdoor activities both groups took part in were also looked at.

## 2.2 TIME OF DATA COLLECTION
Data was collected between November 2017 and April 2018.

## 2.3 SAMPLING, SAMPLE AND DATA COLLECTION
The participants consisted of an opportunity sample of individuals who owned or worked on allotments and a sample of participants who did not spend much time in nature or work on an allotment. These participants were recruited through online gardening forums and nature forums as well as through social media to reach those who did not necessarily spent a large amount of time in nature. There were no limits to where the participants lived as long as they could understand English in order to complete the online survey.

The original sample size was 522 individuals (145 men, 375 women and two participants who preferred to not specify). There were 380 allotment owners and 142 non-allotment owners. Before the main analysis, outliers were removed when it was clear that they were through incorrect data entry. The final data set consisted of 515 participants, 143 men (27.8%), 370 women (71.8%) and two individuals who preferred not to say (0.4%). The final allotment owner sample consisted of 375 individuals (M = 117, F = 257, prefer not to say = 1) and the number of non-allotment owners was 140 individuals (M = 26, F = 113, prefer not to say = 1). Out of those who did own an allotment, 136 worked on a project or allotment group and 379 did not (including those who did not own allotments). The age range was 18–85 years old, with the mean age being 50.6 (SD = 15.4) years old. All the participants needed to be aged over 18 years old however, there was no upper age limit.

The procedure involved an online survey. When working with an online sample, the online survey ethics had to be adapted slightly. For the informed consent, consent was recorded by ticking a box and unable to proceed unless the box had been ticked. Participants were also reminded of their right to withdraw and presented with a debrief form at the end of the online survey which they were provided with a copy of.

Each participant was told they would be taking part in a survey looking at time spent in nature and its effects on mental wellbeing. The participants were given the information sheet and informed consent form. They were then presented with the demographic and individual questions. After they had completed these, they were presented with the main part of the survey consisting of scales on: mental health, self-esteem, physical health, connection to nature, social identity, loneliness, social

support and self-efficacy (appendix 1). Standardised instructions were used for the scales. The participants were then thanked and debriefed. Contact details of relevant organisations were listed in case the participant wished to seek support if needed.

## 2.4 MATERIALS/SURVEY INSTRUMENTS
The materials included a set of scales used to measure: mental health, self-esteem, physical health, connection to nature, social identity, loneliness, social support and self-efficacy. The scales used were the following for mental health (Clarke et al., 2011; NHS health scotland,2008), self-esteem (Robins, Hendin & Trzesniewski, 2001a,b), physical health (DeSalvo et al., 2006), connection to nature (Nisbet & Zelenski, 2013a,b), social identity (Sani, Madhok, Norbury, Dugard & Wakefield, 2014), loneliness (De Jong Gierveld & Van Tilburg, 2006), social support (Van Dick & Haslam, 2012) and self- efficacy (Tambs & Røysamb, 2014). Demographic and individual questions were also added to the survey, all of which were open ended questions to allow for expression and were anonymised where required. These questions involved asking their age, gender, nationality, hours spent in nature per week, whether they owned an allotment and if they did, asking if they took part in a community project or group at the allotments. There was also the question "What activities in nature do you participate in? (For example, walks or gardening, please list as many as apply to you:" This was included as an open-ended question to allow for expression. During analysis any answers of gardening or allotment work were grouped under gardening activities for clarity. The survey was piloted before the main data collection to ensure reliability. The following established scales were used:

### Mental Health – Warwick-Edinburgh – 7 item short version
The first scale was looking at mental health and was the Warwick-Edinburgh short wellbeing scale (NHS Health Scotland, University of Warwick and University of Edinburgh, 2008), a 7 question, 5-point scale (1 = none of the time, 5 = all of the time) assessing mental health through questions such as: "I've been feeling optimistic about the future". The range of scores was 7–35 with a high total indicating good mental health. The scale has been found to be internally consistent and with the same Cronbach's alpha as the WEMWBS (cronbach's alpha range .85–.88). Validity for the scale showed good construct validity (Clarke et al., 2011).

### Self-esteem- Robins et al – Single Item
This scale looked at self-esteem (Robins, Hendin & Trzesniewski, 2001a), using a single item measure on a 7-point scale (1 = not very true of me, 7 = very true of me). The question stated "I have high self-esteem" with participants then asked to rate. The range of scores was

1–7 with a high score indicating high self-esteem. The scale has been found to be internally consistent, although Cronbach's alpha cannot be generated for a single item measure, Heise estimate showed a reliability of .75. Validity was also good with a concurrent validity measure of .72–.76 (Robins, Hendin & Trezesniewski, 2001b).

### Physical Health – general self-rated health comparative

The physical health measure contained two single items, a general health "in general, how would you say your health is….?"and a comparative general health question "Compared to others your age, would you say your health is…?" (DeSalvo et al., 2006). The items used a 5-point scale, from excellent to poor, the range of scores for both measures was 2–10 with lower scores (in the excellent or very good region) indicating good physical health. The measures have been shown to be reliable,with intraclass correlation coefficient of .69 for the general health question and .85 for the comparative one. The scale also showed strong discriminant and concurrent validity (DeSalvo et al., 2006).

### Connection to Nature – Relatedness to Nature Scale Short Form

This scale looked at a person's connection to nature and was the relatedness to nature short form scale (Nisbet & Zelenski, 2013a), using a 6-item measure on a 5-point scale (1 = disagree strongly, 5 = agree strongly). The scale included statements such as "I always think about how my actions affect the environment." The range of scores was 6–30 with a high score indicating connection to nature. The scale has been found to be internally consistent (cronbach's alpha range .60–.86). Validity was good for the scale with convergent validity between .64 and .75 (Nisbet and Zelenski, 2013b).

### Social Identity – Sani Group Identification Scale

This scale was looking at social identification and was the Sani Group Identification Scale (Sani, Madhok, Norbury, Dugard & Wakefield, 2014), a 4 item, 7-point scale (1 = I strongly disagree, 7 = I strongly agree. Questions included: "I feel similar to the other members of my (group)". The range of scores was 4–28 with high scores indicating strong social identity. The scale has been found to be internally consistent (cronbach's alpha range .85–.92). validity for the scale was strong having good levels of stability.

### Loneliness – de Jong Short Form

This scale exploring social loneliness was the de Jong Short Form scale (De Jong Giervekd & Van Tilburg, 2006). This was a six-item, 3-point scale (1 = yes, 3 = no) with some items being reverse coded. Questions included "I miss having people around". The range of scores was

from 6–18. This short form scale had similar reliability to the longer form (Cronbach's alpha = 0.899, Giraldo-Rodríguez, Álvarez-Cisneros & Agudelo-Botero, 2023) as well as adequate validity.

### Social Support – van Dick and Haslam

This scale was looking at Social support and was the van Dick and Haslam scale (van Dick & Haslam, 2012), a 4 question, 7-point scale (1 = not at all, 7 = completely) assessing social support. Questions included: "Do you get the emotional support you need from other people?". The range of scores was 4–28 with a high total indicating good social support. The scale has been found to be internally consistent (Cronbach's alpha range .79–.86). Validity for the scale showed it had strong concurrent and construct validity similar to longer versions of the scale (Jetten, Haslam & Alexander, 2012).

### Self- efficacy – General Self-efficacy short form

This scale was exploring self-efficacy (Tambs & Røysamb, 2014) and was a 5 statement, 4-point scale (1 = not true at all, 4 = exactly true. Statements included "I can always manage to solve difficult problems f I try hard enough'. The range of scores was 5–20 with a high score indicating higher levels of self-efficacy. The scale has been found to be internally consistent (Cronbach's alpha range .79–.88) and has good levels of validity (Romppel et al., 2013).

The dataset has been checked for errors and outliers removed prior to publishing. Also items that needed to (Question 2, 3 and 5 from the loneliness scale) have been reverse coded already.

## 2.5 DATA ANONYMISATION AND ETHICAL ISSUES

Full ethical clearance was given by Nottingham Trent University ethics board. Informed consent was gained from participants before they took part in the survey. Any identifiable data was removed from the survey to ensure confidentiality and anonymity including from the open-ended questions. Participants were also debriefed after the survey.

## 2.6 EXISTING USE OF DATA

The data has been used to test the hypotheses of the study however research has not been published elsewhere.

## (3) DATASET DESCRIPTION AND ACCESS

### 3.1 REPOSITORY LOCATION

Mendeley Data, DOI: 10.17632/8y2h89z46s.3

### 3.2 OBJECT/FILE NAME
Connection to Nature Data

### 3.3 DATA TYPE
Primary data

### 3.4 FORMAT NAMES AND VERSIONS
Excel

### 3.5 LANGUAGE
English

### 3.6 LICENSE
CC BY 4.0

### 3.7 LIMITS TO SHARING
There are no limits to sharing

### 3.8 PUBLICATION DATE
The data was published on 5th August 2024

### 3.9 FAIR DATA/CODEBOOK
The data is easily accessible through a unique identifier and is open for access. The data has used labels which correspond to the sub scales where applicable, access to a file with the coding is available. Instructions are easy to follow and readers have access to the original tools.

## (4) REUSE POTENTIAL

Understanding the importance of time spent in nature and its impacts on wellbeing and mental health is a priority for healthcare and interventions used for mental health. Research is still emerging in this area and research focusing on the use specifically of gardening and allotment style activities compared with other nature engagement are rarely investigated together. This is important to enable green social prescriptions and nature-based interventions to develop and be tailored to the individuals who take part in them. The data could be used to promote positive psychology and nature involvement. As much of the previous research has suggested, considerably more exploration is needed to show the positive effects of being connected to the natural environment, through outdoor activities and time in natural places. Research investigating the amount of time people need to spend in nature to see these positive effects also needs to be explored further.

## ADDITIONAL FILES

The additional files for this article can be found as follows:

- **Appendix 1.** Online survey used. DOI: https://doi.org/10.5334/jopd.122.s1

- **Supplementary File.** Coding sheet for dataset. DOI: https://doi.org/10.5334/jopd.122.s2

## ACKNOWLEDGEMENTS

Acknowledgements to Nottingham Trent University and the research ethics community there, for ethical approval for this data, as this was carried out during an approved psychology program at the university.

## COMPETING INTERESTS

The authors have no competing interests to declare.

## AUTHOR AFFILIATIONS

**Leanne Haywood** orcid.org/0000-0003-4090-3258
University of Chichester, UK
**James Stiller** orcid.org/0000-0002-6122-5911
University of Chichester, UK

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

## PEER REVIEW COMMENTS

*Journal of Open Psychology Data* has blind peer review, which is unblinded upon article acceptance. The editorial history of this article can be downloaded here:

- **PR File 1.** Peer Review History. DOI: https://doi.org/10.5334/jopd.122.pr1

**TO CITE THIS ARTICLE:**
Haywood, L., & Stiller, J. (2024). Data on Connection With the Natural Environment and Its Impact on Mental Health Among Allotment and Non-Allotment Owners. *Journal of Open Psychology Data,* 12: 9, pp. 1–7. https://doi.org/10.5334/jopd.122

**Submitted:** 06 August 2024    **Accepted:** 03 September 2024    **Published:** 13 September 2024

