## [Peer Review History. · Journal of Open Psychology Data]

Response to Reviewer

Thank you for taking the time to review the dataset. I have taken on board the comments and suggestions made and have made the necessary revisions. I will take each comment in turn below and discuss each of the points made, detailing the changes I have made as a result. This can also be seen on the tracked changes made on the paper.

Introduction

In the first paragraph of the Rationale section, the introduction of physical health and physical activity is a little repetitive across a few sentences - consider shortening to make this clearer and concise. Also consider splitting some of the sentences, as a I found it a little less easy to follow the longer ones.

The introduction has been amended to flow better as well as removing repetitive statements to allow ease of reading and understanding. Some items have been rephrased to make clearer statements to the reader as well.

It is also not quite clear whether the two research questions you present at the end are the ones overarching the study (since the sentence before them reads like you are introducing them as an add on "Finally,...")

The aims and main research questions have been edited to make it clear that they are the ones overarching the study and the use of finally has been removed to also help with this.

Methods:

More information on sampling would be helpful: where were the participants from? How were they recruited? Any additional information you can give on the participants would be helpful for researchers who might want to use the data in the future.

More information has been provided in the participants section describing where the participants were recruited from and how they were recruited.

It would be helpful to describe how outliers are removed - what specific criteria did you use?

A mention of outliers being removed due to clear incorrect data input has been added to the methods section to help clear up the reasoning for removal of some data.

It would be helpful for me (and perhaps others) if you could always describe the various scales in the same order (preferably the one they showed up in the questionnaire, or another reason). That would make it easier to find corresponding information between the different sections

Throughout the whole of the paper the scales are now mentioned in the same order, this has also been amended through the coding document for ease of use to the reader.

In the description of the demographics and gardening questions, it would be helpful to note the format of the items. Especially the activity one "Also a question was added asking what activities participants regularly took part in, for this gardening and allotment work were both grouped under gardening activities for clarity during analysis. " was unclear to me from the description, both in terms of what the question was and what you mean by

"grouping": is this referring to the analysis? Or to the item? Also make sure to state which items had open responses and which were closed (and if so, what the answer options were).

This section has been removed and rewritten, giving the exact wording of the question and also to clear up why any mention of gardening or allotments were grouped together. This was in reference to the open-ended question, when participants noted 'gardening activities' or 'allotment activities'. During analysis of the types of activities participants took part in when in nature, any mention of these two activities were grouped together as 'gardening activities'. Which questions were open and closed has also been added into the methods to highlight this and also correspondingly added to the coding document.

Similarly, in the established instruments section, make sure to briefly explain how items were aggregated. For example, in the Warwick-Edinburgh scale, you state the range of scores but not how the scores are actually calculated.

The scales descriptions have also been checked to make sure they feature the range of scores, what the high/low totals indicate, reliability and validity and any other important information to note.

For internal reliability state the actual measures used (Cronbach's alpha?)

This has been checked to make sure all scales used in the survey showed the cronbach's alpha scores for the scales (or any equivalents used in the case of the self-esteem measure).

Describe whether anything has been done with the data prior to publishing (e.g., reversing the coding, checking for errors, etc.)

This has been added into the document, describing how the data has been prepared before being published, including removing outliers and the reasons for this removal as well as item reverse coding and error checks.

Code document:

Please add the answer codes to the established instrument scales for completeness

I have amended the data codes document to include the coding for the scales part of the survey to allow for ease of use.

The headers could be revised: What is now "question type" looks like it is the construct for the established scales, and the broad description for the demographics - I would stick with "construct" and adapt this for the demographics too. "Question" could be rephrased as "item" to be a bit more precise.

These titles have now been amended to 'section type' and 'item' to help make this document more user friendly and avoid confusion with the title names.

It would be easier to read/overview if all tables and the same size and layout

The formatting of tables has been amended so that they all flow with the same layouts.

Other:

I wonder whether including the term allotment in the title would be possible, as it seems like an important part of the study and resulting data

The title has been amended to draw focus onto the allotment group as this is the focal point of the dataset.

Consider describing gender as "men and women" instead of "males and females", as the latter may have more connotations with biological sex than gender (though of course they are linked and terminology can vary in different settings and areas)

This has been changed to Men and Women when describing the dataset demographics and the corresponding information on the coding sheet has been amended to match.

Different fonts are used throughout the paper and supplementary materials, align those in case this is not done in the editing process

The paper has now been formatted with all the same font.

The "codes for data" file is called "not for review" when downloaded - just so you know in case it's the wrong version!

I will double check this when reuploading the amended documents

There are some typos throughout (e.g., "health (DeSalvo et al, 2006).", "whether they owned and allotment and if they did", "The measures have been shown to be reliable. With intraclass correlation coefficient of .69 for the general health question and .85 for the comparative one.")

The paper has been proofread and amendments made where needed in response to this.

The deposited data

The deposited data must include a version that is in an open, non-proprietary format. A csv version should be provided, as excel is a proprietary format. Similarly the Word document containing could be given as a txt file or ODT

The published data set has now been updated providing a txt file for the updated coding document as well as a csv version of the deposited dataset.

d. Studies involving human subjects should adhere to local ethical standards at the host institution and follow American Psychological Association's (APA) Ethical Principles of Psychologists and Code of Conduct (<http://www.apa.org/ethics/code/index.aspx>). Participant data should be sufficiently anonymized and appropriate consent forms should be signed.

Since open responses were used in the questionnaire, it would be good to state in the paper and data description whether these have been anonymised if required.

This has been highlighted in the ethical considerations section of the revised paper.